# Sensitivity and specificity of high-resolution wide field fundus imaging for detecting neovascular age-related macular degeneration

**Maiko Maruyama-Inoue**[1]*, **Yoko Kitajima**[2], **Shaheeda Mohamed**[3], **Tatsuya Inoue**[1], **Shimpei Sato**[2], **Arisa Ito**[1], **Shin Yamane**[1], **Kazuaki Kadonosono**[1]

1 Department of Ophthalmology, Yokohama City University Medical Center, Yokohama, Japan,
2 Department of Ophthalmology, Kanto Rosai Hospital, Kawasaki, Japan, 3 Hong Kong Eye Hospital, Department of Ophthalmology & Visual Sciences, The Chinese University of Hong Kong, Hong Kong, China

☯ These authors contributed equally to this work.
* maicoo@urahp.yokohama-cu.ac.jp

**Data Availability Statement:** All relevant data are within the manuscript.

## Abstract

### Purpose

Early detection and treatment are important management strategies for neovascular age-related macular degeneration (AMD). The purpose of this study was to determine the sensitivity and specificity in detecting neovascular AMD using two wide-field imaging systems: Clarus™ (CLARUS 500™, Carl Zeiss Meditec AG, Jena, Germany) and Optos®(Optos California®, Optos PLC, Dunfermline, United Kingdom), compared to conventional digital fundus photographs.

### Methods

We retrospectively analyzed 109 eyes of 73 consecutive patients with neovascular AMD, who underwent standard examination and multimodal imaging, including fundus photography, and optical coherence tomography (OCT). Unmasked graders utilized slit-lamp biomicroscopy and OCT to diagnose neovascular AMD. Masked graders evaluated Clarus™, Optos®, and digital fundus photograph methods to determine the presence of choroidal neovascularization associated with AMD. Sensitivity and specificity analyses were performed using combined fundoscopy and OCT as the reference standard.

### Results

Ninety eyes were diagnosed with neovascular AMD and the remaining 19 eyes were normal based on the reference standard. Of these, neovascular AMD was detected using Clarus™ in 94.4% (85/90). The sensitivities of Optos® and digital fundus photographs were 81.1% (73/90) and 87.8% (79/90), respectively. The specificities using Clarus™, Optos®, and digital fundus photographs were 89.5% (17/19), 94.7% (18/19), and 89.5% (17/19), respectively.

**Funding:** The authors received no specific funding for this work.

**Competing interests:** The authors have declared that no competing interests exist.

## Conclusion

Clarus™, with its ability to image high-resolution wide field fundus, was considered superior for diagnosing neovascular AMD with high sensitivity and specificity. It may be a useful screening tool for early detection of neovascular AMD, facilitating prompt referral and treatment.

## Introduction

Age-related macular degeneration (AMD) is the leading cause of severe irreversible vision loss in older adults in the United States and other developed countries [1, 2]. Recently, anti-vascular endothelial growth factor (VEGF) agents have demonstrated efficacy in improving visual acuity outcomes in patients with neovascular AMD [3, 4]. However, larger lesion size and worse visual acuity at baseline were associated with a poorer visual outcome [5]. Therefore, early detection and treatment initiation at onset of neovascular AMD results in better therapeutic outcomes [6]. In this regard, there are several fundus imaging systems available for screening. Clarus™ (CLARUS 500™, Carl Zeiss Meditec AG, Jena, Germany) is a newly designed scanning laser ophthalmoscope that can obtain 133-degree field of the retina in a single image without mydriasis. Also incorporated into this system are features of partially confocal optics and true color imaging using red, green, and blue laser ophthalmoscopy scans. The high-resolution of 7.3 microns and high image quality that avoids eyelash and eyelid artifacts enables diagnosis of a variety of lesions in the retina.

Optos® (Optos California®, Optos PLC, Dunfermline, United Kingdom) also consists of an ultra-wide field (UWF) image that can obtain 200-degree field of the retina in a single image without mydriasis. However, the Optos® fundus image is slightly different from a real color image because it combines monochromatic red and green scanning laser ophthalmoscopy scans. Optos® has been useful for detecting a variety of retinal lesions such as retinal tears, retinal detachment, and diabetic retinopathy (DR) or peripheral macular lesions [7–9]. However, eyelash artifacts with Optos® may prevent clear imaging of the inferior periphery [8].

Digital fundus photograph (TRC-50DX, Topcon, Tokyo, Japan), a mydriatic high-resolution fundus photography with retinal images of 35 or 50-degree field, is frequently used for detecting macular lesions [10, 11] However, this method is easily influenced by artifacts such as cataract and small pupils [12].

The purpose of this study was to evaluate the accuracy of Clarus™, Optos®, and digital fundus photographs for detecting neovascular AMD. The sensitivity and specificity of these techniques were determined by comparing to the reference standard, which included fundoscopy combined with optical coherence tomography (OCT) data.

## Patients and methods

### Study design

Between June and July 2019, Japanese patients diagnosed with neovascular AMD who were seen in the Department of Ophthalmology at the Yokohama City University Medical Center formed the study population. The medical records of 73 (146 eyes) consecutive patients were extracted for this retrospective analysis. The study was conducted according to the principles of the Declaration of Helsinki, and informed consent was obtained from all eligible patients. This study was approved by the Ethics Committee of the Yokohama City University Medical Center.

## Data collection

All patients underwent ophthalmologic examination, including slit-lamp biomicroscopy, spectral-domain OCT (SD-OCT) imaging (Heidelberg Spectralis HRA + OCT; Heidelberg Engineering, Germany), and color fundus photography using Clarus™, Optos®, and digital fundus photographs. Images obtained by Clarus™ were taken in a single-shot of 133-degree field or auto-montaged image of 200-degree field with mydriasis. Images obtained by Optos® and the digital fundus photographs were from a single-shot with mydriasis. Images of 50-degree field of macula in each imaging device were used for evaluation by masked graders.

## Unmasked evaluation of neovascular AMD

The patients were examined by two unmasked graders (MM and YK). The findings obtained from slit-lamp biomicroscopy and SD-OCT imaging in the macula were reviewed by the unmasked graders. We included their fellow eyes with neovascular AMD and normal fundus. However, eyes that showed age-related maculopathy (ARM) were excluded. Normal fundus was defined as none or minimal macular changes of age-related disease. Lesions in ARM can be early with drusen and/or mild retinal pigment epithelium abnormalities or late with features of geographic atrophy [13]. Neovascular AMD was defined as "present" if there were any characteristic signs of choroidal neovascularization (CNV) on medical examination. Both active and inactive CNV in the diagnosis of neovascular AMD were included in this study.

Of the 73 patients, 17 patients had bilateral neovascular AMD, 37 fellow eyes showed ARM, and the remaining 19 fellow eyes were graded as normal by the unmasked graders. Therefore, the fundus images of a consecutive series of 109 eyes of 73 patients (both treatment naïve and treated eyes) with neovascular AMD were retrospectively reviewed by the masked graders.

## Masked evaluation of sensitivity and specificity of retinal imaging

Two retina specialists (SS and AI) who were blinded to the purpose and results of the study evaluated images from the three modalities used in this study—Clarus™, Optos®, and digital fundus photographs, without any additional patient information. Grading for neovascular AMD was performed in a binary manner (1 = present, 0 = absent). The graders were allowed to adjust magnification and evaluated 50-degree field of macula. Retinal images thus obtained were evaluated two times after an interval of one week. When the evaluation was inconsistent, a third masked reader (TI) made the final arbitration. By using the first obtained data from the masked graders, sensitivity and specificity of the three retinal imaging devices for diagnosing neovascular AMD were determined by comparison against a reference, the analysis using combined slit-lamp biomicroscopy and SD-OCT information by unmasked graders. Figs 1 and 2 show example images of neovascular AMD and normal eyes, respectively.

## Statistical analysis

The two graders' inter-observer and intra-observer agreements were assessed using the kappa statistic. Inter-observer agreements were assessed by using the first obtained data from the masked graders. Intra-observer agreements were assessed by using the data obtained two times after an interval of one week. Kappa statistic was defined as follows: greater than 0.81 represents "excellent" agreement; 0.61–0.80 represents "good" agreement; 0.41–0.60 represents "moderate" agreement and less than 0.40 represents "poor" agreement [14]. The sensitivity and specificity of the fundus imaging systems were compared using McNemar test. Proportion of phakic eyes in true positives/negatives or false positives/negatives was compared

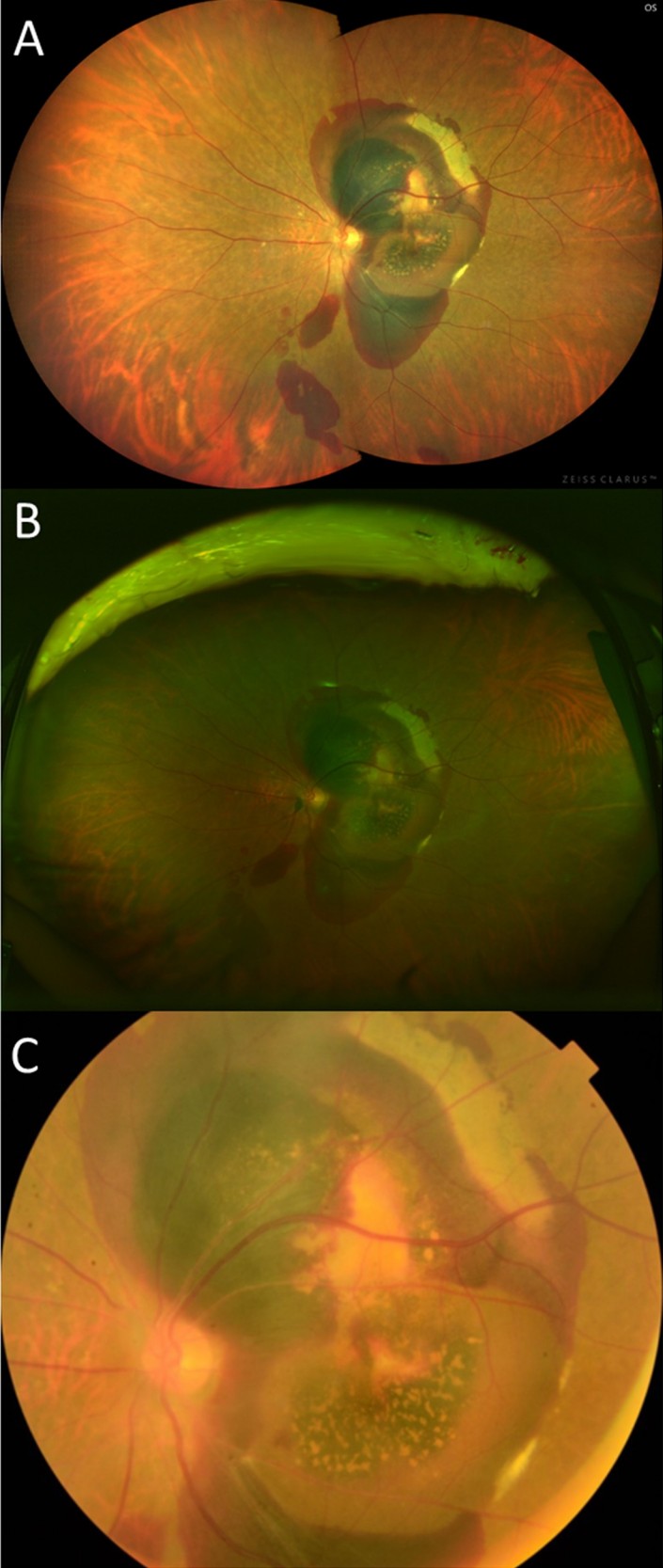

**Fig 1. Fundus photographs of an eye with neovascular AMD obtained using the three imaging systems.** (A) Clarus™; (B) Optos®; (C) Digital fundus photograph.

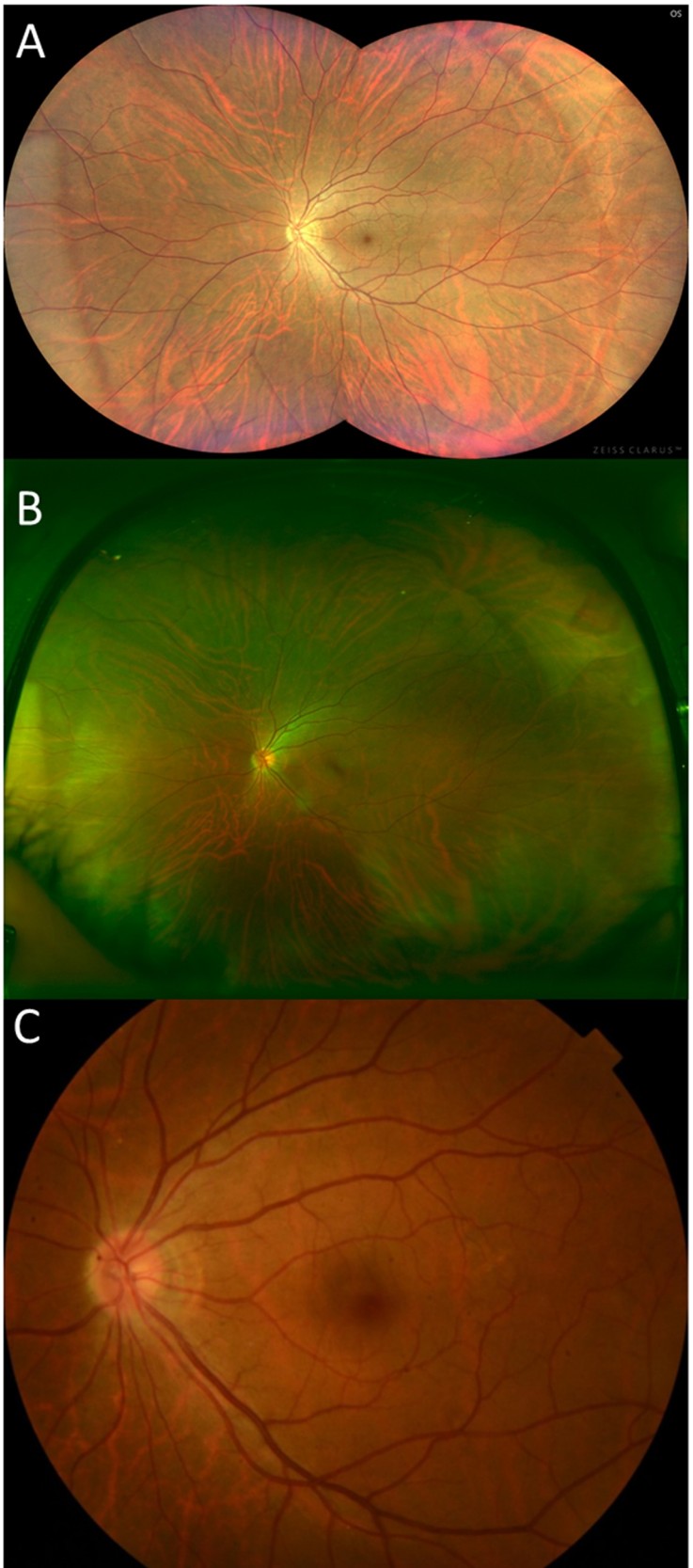

**Fig 2. Fundus photographs of a normal eye obtained using the three imaging systems.** (A) Clarus™; (B) Optos®; (C) Digital fundus photograph.

**Table 1. Clinical characteristics of the patients with neovascular AMD.**

| | |
|---|---|
| Number of patients | 73 |
| Number of eyes | 109 |
| Age, mean ± SD, year | 77.5 ± 8.4 |
| Sex(Male/Female) | 52/21 |
| Lens status(Phakic/Pseudophakic) | 73/36 |
| Baseline logMAR visual acuity, mean ± SD | |
| Eyes with neovascular AMD | 0.423 ± 0.496 (Snellen equivalent 20/53) |
| Eyes with normal | -0.043 ± 0.062 (Snellen equivalent 20/18) |

AMD = age-related macular degeneration; SD = standard deviation; logMAR = logarithm of the minimum angle of resolution.

using Fisher's exact test. Statistical analysis was performed using Ekuseru-Toukei 2012 (Social Survey Research Information, Tokyo, Japan). A *P* value <0.05 was considered statistically significant.

## Results

### Patient characteristics and unmasked evaluation of neovascular AMD

Demographic and clinical characteristics of patients with AMD are shown in Table 1. Of the 109 eyes, 90 (82.6%) exhibited neovascular AMD and the remaining 19 eyes (17.4%) were marked normal. Mean patient age was 77.5 ± 8.4 years (median, 78 years; range, 57–97 years). Of the 73 patients, 52 were men and 21 were women. Among the 109 eye, 73 eyes were phakic and 36 eyes were pseudophakic. Mean logMAR best corrected visual acuity was 0.423 (20/53 Snellen equivalent) in the affected eyes and −0.043 (20/18 Snellen equivalent) in the fellow normal eyes.

### Inter- and intra- grader agreement for diagnosing AMD

Inter-observer agreement showed a good kappa value (± standard error) of 0.640 ± 0.083 ($P < 0.001$) for Clarus™, moderate kappa of 0.595 ± 0.075 ($P < 0.001$) for Optos®, and 0.543 ± 0.080 ($P < 0.001$) for the digital fundus photographs (Table 2).

Intra-observer agreement showed an excellent kappa of 0.882 ± 0.058 ($P < 0.001$) for Clarus™, 0.810 ± 0.069 ($P < 0.001$) for Optos®, and 0.878 ± 0.060 ($P < 0.001$) for the digital fundus photographs for grader 1 (SS, Table 3). Intra-observer agreement showed an excellent kappa of 0.863 ± 0.054 ($P < 0.001$) for Clarus™, good kappa of 0.743 ± 0.065 (P < 0.001) for

**Table 2. Inter-observer agreement for detecting CNV in three imaging modalities.**

| | Grader 1 | Grader 2 | | Agreement (%) | Kappa value | *P* value |
|---|---|---|---|---|---|---|
| | | + | - | | | |
| Clarus™ | + | 74 | 13 | 86.2 | 0.640 | P < 0.001 |
| | - | 2 | 20 | | | |
| Optos® | + | 64 | 20 | 81.7 | 0.595 | P < 0.001 |
| | - | 0 | 25 | | | |
| Digital fundus photograph | + | 68 | 21 | 80.7 | 0.543 | P < 0.001 |
| | - | 0 | 20 | | | |

**Table 3. Intra-observer agreement for detecting CNV in three imaging modalities (grader 1).**

| | First data | Second data | | Agreement (%) | Kappa value | *P* value |
|---|---|---|---|---|---|---|
| | | + | - | | | |
| Clarus™ | + | 86 | 1 | 96.3 | 0.882 | *P* < 0.001 |
| | - | 3 | 19 | | | |
| Optos® | + | 82 | 2 | 93.6 | 0.810 | *P* < 0.001 |
| | - | 5 | 20 | | | |
| Digital fundus photograph | + | 87 | 2 | 96.3 | 0.878 | *P* < 0.001 |
| | - | 2 | 18 | | | |

Optos®, and 0.795 ± 0.061 (*P* < 0.001) for the digital fundus photographs for grader 2 (AI, Table 4).

## Masked grading outcomes and analyses of accuracy

A diagnosis of neovascular AMD was confirmed using Clarus™ in 85 eyes, Optos® in 73 eyes, and digital fundus photographs in 79 eyes by the masked readers. On the other hand, a diagnosis of normal fundus was confirmed using Clarus™ in 17 eyes, Optos® in 18 eyes, and digital fundus photographs in 17 eyes. The sensitivities for the detection of neovascular AMD by Clarus™, Optos®, and digital fundus photographs were 94.4% (85/90), 81.1% (73/90), and 87.8% (79/90), respectively, compared with the reference. The specificities were 89.5% (17/19), 94.7% (18/19), and 89.5% (17/19), respectively (Table 5). The sensitivity of the Clarus™ was significantly higher than Optos® (*P* = 0.010), but not significantly higher than digital fundus photographs (*P* = 0.211). The sensitivity of the digital fundus photographs was not also significantly higher than Optos® (*P* = 0.211). On the other hand, the specificity of the fundus imaging systems was not significantly different (all *P* > 0.05).

## Sub-analyses in phakic versus pseudophakic eyes

Of the 109 eyes, false positives or false negatives were recorded using Clarus™ in 7 eyes, Optos® in 18 eyes, and digital fundus photographs in 13 eyes. All 7 eyes which showed false positives/negatives using Clarus™ were phakic eyes. Of the 18 eyes which showed false positives/negatives using Optos®, 15 eyes were phakic and remaining 3 eyes were pseudophakic. Among the 13 eyes which had false positives/negatives using the digital fundus photographs, 10 eyes were phakic and remaining 3 eyes were pseudophakic. Proportion of phakic or pseudophakic in each imaging modalities is shown in Table 6. Although there were no significant differences in three imaging modalities, Clarus™ tended to have less proportion of phakic eyes

**Table 4. Intra-observer agreement for detecting CNV in three imaging modalities (grader 2).**

| | First data | Second data | | Agreement(%) | Kappa value | *P* value |
|---|---|---|---|---|---|---|
| | | + | - | | | |
| Clarus™ | + | 76 | 0 | 94.5 | 0.863 | *P* < 0.001 |
| | - | 6 | 27 | | | |
| Optos® | + | 64 | 0 | 88.1 | 0.743 | *P* < 0.001 |
| | - | 13 | 32 | | | |
| Digital fundus photograph | + | 68 | 0 | 90.8 | 0.795 | *P* < 0.001 |
| | - | 10 | 31 | | | |

**Table 5. Sensitivity and specificity for diagnosing neovascular AMD in three imaging modalities.**

|  | Unmasked graders | Masked graders |  | Sensitivity (%) | Specificity (%) |
|---|---|---|---|---|---|
|  |  | + | - |  |  |
| Clarus™ | + | 85 | 5 | 94.4 | 89.5 |
|  | - | 2 | 17 |  |  |
| Optos® | + | 73 | 17 | 81.1 | 94.7 |
|  | - | 1 | 18 |  |  |
| Digital fundus photograph | + | 79 | 11 | 87.8 | 89.5 |
|  | - | 2 | 17 |  |  |

which showed false positives/negatives. Also, true positives/negatives were recorded in all pseudophakic eyes using the Clarus™ (Table 6).

## Discussion

In this study, three types of retinal imaging systems, namely, Clarus™, Optos®, and digital fundus photographs, were used to detect neovascular AMD. Clarus™ had the highest sensitivity (94.4%) for diagnosing neovascular AMD, compared to Optos® and digital fundus photographs. This is the first report that describes the usefulness of Clarus™ for detecting neovascular AMD.

It has been estimated that worldwide, 8.7% of the general population suffers from AMD, and there is an upward trend in the projected number of patients in the next two decades [15]. Recent developments in treatments include the use of inhibitors of VEGF for blocking CNV in patients with neovascular AMD [3, 4]. Therefore, early detection and diagnosis of neovascular AMD enables prompt treatment for the maximum benefit. The current standard for diagnosing neovascular AMD relies on fluorescein angiography and OCT [16, 17]. However, fundus camera has been extensively used and is still important for early detection and also for examining eyes with neovascular AMD [10] and evaluating lesions before and after treatment.

Recently, an UWF imaging has been widely used for screening or identification of retinal pathology [18]. For example, Optos UWF images improved detection of peripheral lesions in DR and enabled more accurate classification of the disease [19]. Hirano et al compared Clarus™ with Optos® in patients with DR and described that both systems were useful for assessing DR severity [20]. However, the comparison of wide-field imaging systems with conventional digital fundus camera for diagnosing macular lesions, especially neovascular AMD, remains unclear.

In this study, two retina specialists, who had ophthalmology residency for more than 5 years and retina-vitreous fellowship for more than 2 years, were selected as masked graders with similar abilities as unmasked graders. The intra- and inter-observer agreements for three

**Table 6. Proportion of phakic/pseudophakic in three imaging modalities.**

|  |  | Phakic (%) | Pseudophakic (%) | P-value |
|---|---|---|---|---|
| Clarus™ | True positives/negatives | 66(60.6) | 36(33.0) | P = 0.093 |
|  | False positives/negatives | 7(6.4) | 0(0) |  |
| Optos® | True positives/negatives | 58(53.2) | 33(30.3) | P = 0.169 |
|  | False positives/negatives | 15(13.8) | 3(2.7) |  |
| Digital fundus photograph | True positives/negatives | 63(57.8) | 33(30.3) | P = 0.539 |
|  | False positives/negatives | 10(9.2) | 3(2.7) |  |

retinal imaging systems were moderate or good, suggesting a high degree of repeatability and reproducibility.

We found that the sensitivity for detecting neovascular AMD using Clarus™ was 94.4%. In comparison, the sensitivities of Optos® and digital fundus photographs were 81.1% and 87.8%, respectively. The sensitivity of Clarus™ was significantly higher than Optos®. We speculated that lesions associated with neovascular AMD such as hemorrhage or fluid can be represented more clearly using true color imaging by red, green, and blue laser ophthalmology scans incorporated in Clarus™. Furthermore, Clarus™ has a wide-field retinal camera with a 7-micron resolution that detects retinal lesions even in the macula. In addition to its partially confocal optics, because red scanning laser ophthalmoscopy penetrates deep into the choroid, Clarus™ may not be easily affected by cataracts (Fig 3). In fact, Clarus™ tended to have the

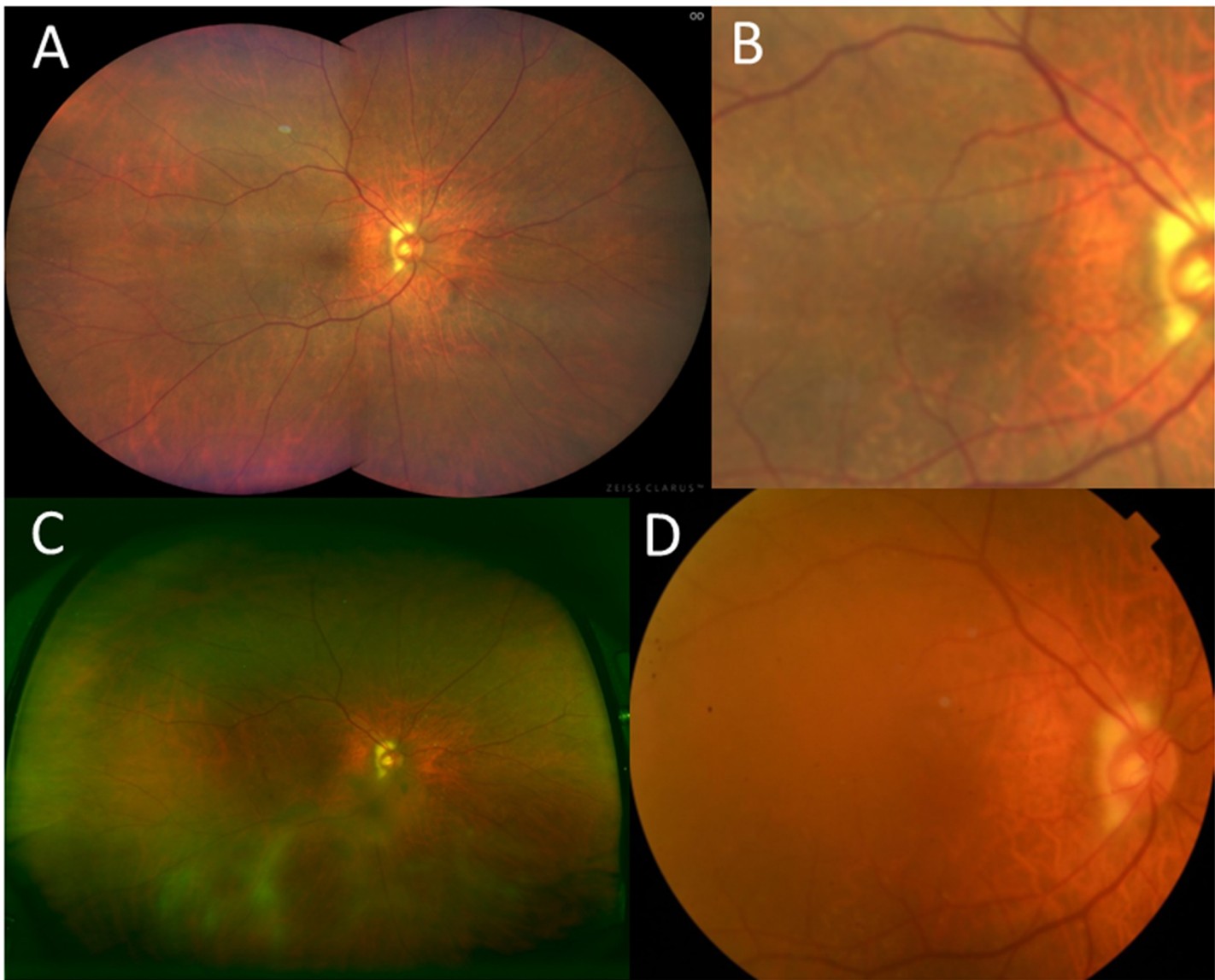

**Fig 3. Fundus photographs of a normal eye obtained using the three imaging systems.** (A) Clear imaging obtained by Clarus™; (B) Clarus™ imaging with 50-degree field of macula. (C) Blurred imaging obtained by Optos®, particularly in the inferior area; (D) Imaging obtained by the digital fundus photograph is obscured due to cataract.

least proportion of phakic eyes which showed false positives/negatives in three imaging devices. Optos® offers a combination of monochromatic red and green scanning laser ophthalmocsopy, which may prevent clear imaging of retinal lesions (Fig 4); eyelash artifact or vitreous opacity may also obscure the fundus photographs [8]. Also, the resolution of Optos® is 14 microns, which is lower than that of Clarus™, affecting accurate imaging of the macula.

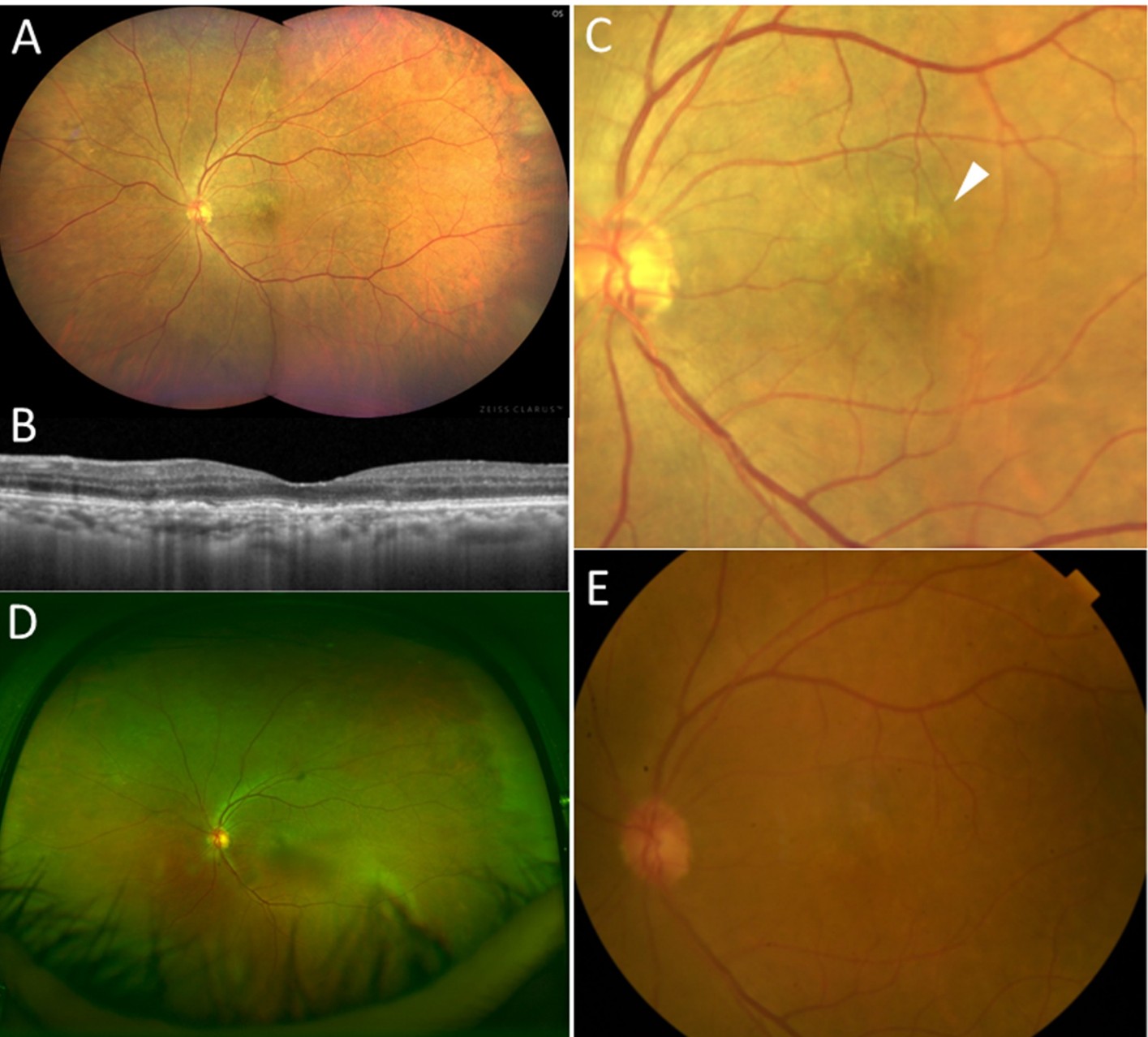

**Fig 4. Fundus photographs of an eye with neovascular AMD which was diagnosed as normal, using Optos® and digital fundus photograph.** (A) Imaging obtained by Clarus™ (B) OCT imaging shows inactive occult CNV. (C) Clarus™ imaging with 50-degree field of macula shows the area of retinal pigment epithelium (RPE) alteration in the macula (white arrowhead); two masked graders classified the image as neovascular AMD; (D) Optos® imaging shows unclear RPE alteration; one masked grader classified the image as neovascular AMD while the other grader and the third grader also diagnosed it as normal; (E) Digital fundus photograph shows obscure RPE alteration; one masked grader classified the image as neovascular AMD while the other grader and the third grader also diagnosed it as normal.

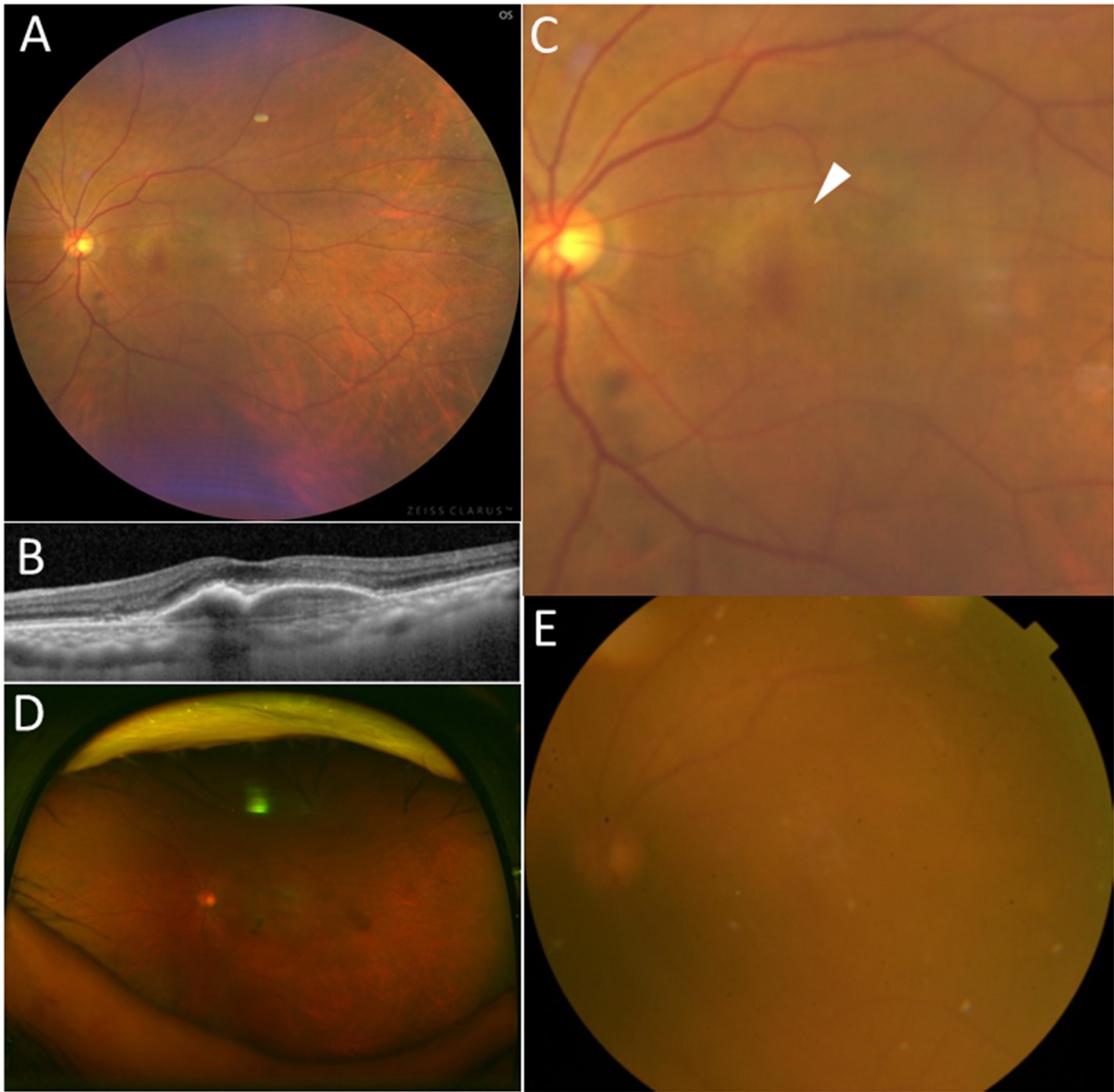

**Fig 5. Fundus photographs of an eye with neovascular AMD, which was diagnosed as normal using Optos® and digital fundus photograph.** (A) Imaging obtained by Clarus™ (C) OCT imaging shows active fibrovascular pigment epithelium detachment. (B) Clarus™ imaging with 50-degree field of macula shows a neovascular AMD lesion (white arrowhead); two masked graders classified the image as neovascular AMD; (D) Optos® imaging shows an unclear lesion that was classified as normal by two masked graders; (E) Digital fundus photograph shows unclear imaging due to cataract; two masked graders also classified the image as normal.

Although the sensitivity of Optos® tended to be lower than digital fundus photographs, there were no significant differences between the two modalities, which was consistent with a previous report that there was a good agreement between grading by digital fundus camera and Optos® in the macula [21].

On the other hand, although the sensitivity showed no significant differences between Clarus™ and digital fundus photographs, digital fundus photographs tended to have lower sensitivity than Clarus™. Digital fundus photographs depict high-resolution images, however, media opacity such as cataract or small pupil deteriorates image quality (Fig 5) [12]. In this study, the proportion of phakic patients who showed false positives/negatives using digital fundus photographs tended to be higher than that of Clarus™. High detection sensitivity achieved by Clarus™ suggests that it may be a useful tool for the clinical diagnosis of neovascular AMD, and potentially, the reduced need for slit-lamp examination in the future. For example, the Clarus™ can be potentially beneficial when realizing telemedicine systems, which could help to streamline the AMD referral process, reduce waiting times and reduce the overall burden of healthcare costs.

All three imaging systems evaluated in this study showed high specificity, with few false positives, indicating that these imaging modalities are likely to result only in necessary referrals.

In this study, the male ratio was much higher than that of female. In Japan, patients with neovascular AMD have a male predominance [22], which is converse to population-based studies in Caucasians [23, 24]. Although the reason for this is unclear, genetic differences between Japanese and Caucasians or the higher smoking rate in males in Japan may underlie such differences.

The limitations of this study are its retrospective nature and the small sample size. The results of this study need to be validated with further prospective studies involving more patients. Although Clarus™ and Optos® can be usually taken without mydriasis, in this study, all images were taken with mydriasis. Others have demonstrated that mydriatic image obtained using the Optos® system is better than non-mydriatic images for determining severity of DR [25]. Therefore, it will be important to investigate the reliability of Clarus™ to discern in the presence or absence of mydriasis. Because of the high risk for progression to AMD, early detection for ARM is also helpful to initiate treatment earlier. However, we excluded eyes with ARM in this study, and future investigations should consider the sensitivity and specificity of Clarus™ for detecting ARM.

## Conclusion

Clarus™, with its ability to image high-resolution wide field fundus, was considered superior for diagnosing neovascular AMD with high sensitivity and specificity. It may be a useful screening tool for early detection of neovascular AMD, facilitating prompt referral and treatment.

## Author Contributions

**Conceptualization:** Maiko Maruyama-Inoue, Tatsuya Inoue, Shin Yamane.

**Data curation:** Maiko Maruyama-Inoue, Yoko Kitajima.

**Investigation:** Maiko Maruyama-Inoue, Yoko Kitajima, Tatsuya Inoue, Shimpei Sato, Arisa Ito.

**Supervision:** Kazuaki Kadonosono.

**Writing – original draft:** Maiko Maruyama-Inoue.

**Writing – review & editing:** Maiko Maruyama-Inoue, Shaheeda Mohamed.

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
