## [Decision Letter · Decision Letter 0]

20 May 2020

PONE-D-20-09112

Sensitivity and Specificity of High-resolution Wide Field Fundus Imaging for Detecting Age-related Macular Degeneration

PLOS ONE

Dear Dr Maruyama-Inoue,

Thank you for submitting your manuscript to PLOS ONE. After careful consideration, we feel that it has merit but does not fully meet PLOS ONE’s publication criteria as it currently stands. Therefore, we invite you to submit a revised version of the manuscript that addresses the points raised during the review process.

Both reviewers found significant areas that need to be improved and I agree with them. Please submit a revised version with these into account. 

We would appreciate receiving your revised manuscript by Jul 04 2020 11:59PM. To enhance the reproducibility of your results, we recommend that if applicable you deposit your laboratory protocols in protocols.io, where a protocol can be assigned its own identifier (DOI) such that it can be cited independently in the future. For instructions see: http://journals.plos.org/plosone/s/submission-guidelines#loc-laboratory-protocols

We look forward to receiving your revised manuscript.

Kind regards,

Demetrios G. Vavvas

Academic Editor

PLOS ONE

Journal Requirements:

Reviewers' comments:

Reviewer's Responses to Questions

**Comments to the Author**

1. Is the manuscript technically sound, and do the data support the conclusions?

Reviewer #1: Partly

Reviewer #2: Yes

2. Has the statistical analysis been performed appropriately and rigorously? 

Reviewer #1: No

Reviewer #2: N/A

3. Have the authors made all data underlying the findings in their manuscript fully available?

Reviewer #1: No

Reviewer #2: Yes

4. Is the manuscript presented in an intelligible fashion and written in standard English?

Reviewer #1: Yes

Reviewer #2: Yes

5. Review Comments to the Author

Reviewer #1: Major revisions required:

1. Clarus images were not taken using universal methodology (either single shot or montage images). Since the whole aim of this study is to compare agreement and sensitivity/specificity of two imaging devices, the imaging protocol for each device has to be universal for all eyes included.

2. Inter-observer agreement is rather low. (barely above 0.6 for clarus and below 0.6 for optos and digital fundus photos ) This essentially means that for the binary task of presence/absence of CNV, graders in this study disagreed in about half of the images. please see relevant literature on inter-observer agreement on CNV. I would suggest graders to go over the images again to attempt reaching agreement at an acceptable for scientific literature level.

3. lines 196-197: ‘moderate or good [agreement] suggesting a high degree of reproducibility and subjectivity’. Please see above comment. Also, ’Subjectivity’ does not seem a positive characteristic in a retinal imaging study.

4. Study population and inclusion criteria is unclear.

(a) Methods lines 146-147 say ‘In this study, unmasked graders evaluated 109 eyes of 73 patients with neovascular AMD’ then lines 149-150 ‘Of the 109 eyes, 90 (82.6%) exhibited neovascular AMD and remaining 19 eyes (17.4%) were marked normal’.

(b) Lines 152-153: were fellow eyes included in this study ? were all those fellow eyes normal while the contralateral eye had nAMD ?

5. The whole discussion has 1 reference (!) - and this is in the limitations part. The total number of 8 references for the whole manuscript is surprisingly low, especially for a retinal imaging study. There is a plethora of imaging articles that should be discussed herein for comparison with the authors results. In the current version there is a marked lack of an elaborate discussion.

Additional revisions required:

6. Lines 206-207: please add phakia status and relevant subgroup analysis in phakic vs pseudophakic eyes since this is discussed as a potential advantage of Clarus over Optos.

7.Title has to be revised since this study only investigated neovascular AMD (nAMD). similar changes have to be made in introduction of abstract and manuscript.

8. Abstract Conclusion : ‘improving therapeutic outcomes and maintenance of better visual acuity’ this is not supported by the study’s results. Similar changes needed in manuscripts conclusion line 264.

9. Abstract Conclusion: ’superior for examination of detailed macular lesions’ is not supported by study’s results. Extensive evaluation of a device in multiple aspects is needed before attempting to claim superiority and this study is limited in terms of methodology. also the authors did not evaluate multiple macular lesions herein.

Reviewer #2: Thank you for submitting your manuscript. While the purpose of this study is of great scientific interest, in my opinion many aspects of your manuscript need to be extensively improved.

Major remarks

The “Introduction” section (lines 56-79) could be significantly strengthened. While I am aware of your scientific work and your writing style, I would strongly suggest you expand this section, review the literature but also add relevant references. In many statements I believe that a reference is missing such as in line 59 but also the 2nd and the 3rd paragraphs have no references at all. I would strongly encourage you to at least site a website in order to support the content when needed. In addition, in the 2nd sentence (line 57) you will also need to clarify that you are actually talking about neovascular AMD. This should also be added in your “Purpose” section (Abstract, first sentence) but I would suggest you add it also in your title, the short title and in the “Discussion” (e.g. lines 88, 190). While these statements might be true, you should support those based on other publications and maybe add some epidemiological data as well. The same applies for line 59 regarding screening. In general, this weakness is found throughout your manuscript.

In regard to your “Data Collection”, you do not really talk about the actual numbers. It seems like most of your patients had bilateral neovascular AMD. This is an important point not only to highlight it in your Methods but also consider it during your analysis and interpretation of the results.

Please, add a citation for ARM (lines 85-97).

Line 107: When you say that “Data obtained from slit-lamp biomicroscopy and SD-OCT were reviewed by two unmasked graders” do you mean that the patient was examined by two ophthalmologists? What kind of data did you collect? Please, explain/describe those characteristics.

Line 149: You say that you tested 109 from 73 patients. Those 19 eyes that were marked normal are not actually “the remaining”. You need to explain whether you scanned in total both eyes of the 73 patients (n=146) and if yes, what happened to the remaining 18 eyes. Did all of these belong to the ARM group? Were they excluded for other reason (e.g scan quality)?

Please, expand your “Introduction” and “Discussion” sections and talk more about the strengths and the weaknesses of these three devices. Also, based on the current literature, make a comment on the use of this device in other retina diseases.

I would include the lens status of those patients not only in the demographic table but also consider it as a parameter in your statistical analysis. You mention that “Clarus may not be easily affected by cataracts” (lines 206-207). If so, it would be interesting to see the results of this analysis.

Please, consult a statistician. Explain why you chose to use McNemar’s test and not Cohen’s.

Also, it would be better to use tables in order to present the results of your analysis.

Minor remarks

Consider adding keywords. This can increase the number of people finding and reading your article.

You could make a comment on the male to female ratio of your cohort compared to other cohorts and studies.

Line 252: please, review the sentence. In general, the quality of your manuscript can be improved even more after a careful proofreading.

Thank you.

6. PLOS authors have the option to publish the peer review history of their article (what does this mean?). If published, this will include your full peer review and any attached files.

Reviewer #1: No

Reviewer #2: No

---

## [Author Response · Author response to Decision Letter 0]

4 Jun 2020

June 2nd, 2020

Demetrios G. Vavvas, MD, PhD. 

Academic Editor

PLOS ONE

RE: PONE-D-20-09112, entitled "Sensitivity and Specificity of High-resolution Wide Field Fundus Imaging for Detecting Age-related Macular Degeneration"

Dear Dr. Demetrios G. Vavvas

Thank you very much for your kind reviewing our manuscript.

You will find the author’s comments for manuscript on this letter which was revised based on the reviewer’s comments.

We are looking forward to seeing your favorable comments in the future.

We greatly appreciate your editing work. 

With Best regards,

Corresponding author;

Maiko Maruyama-Inoue, M.D. 

Yokohama City University Medical Center

4-57 Urafune-cho, Minami-ku, Yokohama, Kanagawa 232-0024, Japan.

E-mail: maicoo@urahp.yokohama-cu.ac.jp

Fax: +81 45 253 8490

Phone: +81 45 261 5656 

Journal Requirements

Response: Thank you for your requirements. I ensured that this manuscript meets PLOS ONE’s style requirements. 

Reviewer’s comments

Reviewer #1

1. Clarus images were not taken using universal methodology (either single shot or montage images). Since the whole aim of this study is to compare agreement and sensitivity/specificity of two imaging devices, the imaging protocol for each device has to be universal for all eyes included.

Response: We thank the reviewer for these comments. As you point out, it is important to be universal for this study. Although Clarus images were taken in a single-shot of 133-degree field or auto-montaged image of 200-degree field, we used 50-degree field of the macula when evaluating. Images taken by Optos and fundus photograph were also evaluated using 50-degree field. We added the sentence about it. 

Page 6, line 106-107

Images of 50-degree field of macula in each imaging devices were used when these images are evaluated for the masked graders.

2. Inter-observer agreement is rather low. (barely above 0.6 for clarus and below 0.6 for optos and digital fundus photos ) This essentially means that for the binary task of presence/absence of CNV, graders in this study disagreed in about half of the images. please see relevant literature on inter-observer agreement on CNV. I would suggest graders to go over the images again to attempt reaching agreement at an acceptable for scientific literature level.

Response: Thank you for the comment. Kappa value is different from the proportion of agreement. Proportion of agreement in Clarus was 94/109 (86.2%, Kappa value was 0.640), in Optos was 89/109 (81.7%, Kappa value was 0.595), and in digital fundus photograph was 88/109 (80.7%, Kappa value was 0.543). We think these values are acceptable for scientific literature level. Our previous report, which compared FA with OCTA for detecting CNV, showed that inter-observer agreement was 0.454-0.686 (Inoue M et al, IOVS, 2016). It was almost the same as this study. 

Table 2

I added the proportion of the agreement.

3. lines 196-197: ‘moderate or good [agreement] suggesting a high degree of reproducibility and subjectivity’. Please see above comment. Also, ’Subjectivity’ does not seem a positive characteristic in a retinal imaging study.

Response: We thank the reviewer for this suggestion. This is our mistake. We deleted the word ‘subjectivity’.

Page 15, line 236-238

The intra- and inter-observer agreements for three retinal imaging systems were moderate or good, suggesting a high degree of repeatability and reproducibility. 

4. Study population and inclusion criteria is unclear.

(a) Methods lines 146-147 say ‘In this study, unmasked graders evaluated 109 eyes of 73 patients with neovascular AMD’ then lines 149-150 ‘Of the 109 eyes, 90 (82.6%) exhibited neovascular AMD and remaining 19 eyes (17.4%) were marked normal’.

(b) Lines 152-153: were fellow eyes included in this study ? were all those fellow eyes normal while the contralateral eye had nAMD ?

Reponse: Thank you for the comments. In Methods section, we described the inclusion criteria. We included the normal fellow eye but excluded eyes which had ARM. We added the sentence in the result section.

Page 7, line 112-116

We included their fellow eyes with neovascualr AMD and normal fundus. However, eyes that showed ARM were excluded. Normal fundus was defined as none or minimal macular changes of age-related disease. Lesions in ARM can be early with drusen and/or mild retinal pigment epithelium abnormalities or late with features of geographic atrophy.

Page 7, line 120-124

Of the 73 patients, 17 patients had bilateral neovascualr AMD, 37 fellow eyes showed age-related maculopathy (ARM), and remaining 19 fellow eyes were normal by the unmasked graders. Therefore, the fundus imaging of a consecutive series of 109 eyes of 73 patients (both treatment naïve and treated eyes) with neovascular AMD were retrospectively reviewed by the masked graders.

5. The whole discussion has 1 reference (!) - and this is in the limitations part. The total number of 8 references for the whole manuscript is surprisingly low, especially for a retinal imaging study. There is a plethora of imaging articles that should be discussed herein for comparison with the authors results. In the current version there is a marked lack of an elaborate discussion.

Response: We thank the reviewer for the comment. We added some paragraphs in the discussion section and finally listed 25 references.　

Additional revisions required:

6. Lines 206-207: please add phakia status and relevant subgroup analysis in phakic vs pseudophakic eyes since this is discussed as a potential advantage of Clarus over Optos.

Response: Thank you for the suggestion. I added subgroup analysis in phakic and pseudophakic eyes and discussed it. 

Page 13, line 199-209 and table 6

Of the 109 eyes, false positives or false negatives were recorded using the ClarusTM in 7 eyes, the Optos® in 18 eyes, and the digital fundus photographs in 13 eyes. All 7 eyes which showed false positives/negatives using the ClarusTM were phakic eyes. Of the 18 eyes which showed false positives/negatives using the Optos®, 15 eyes were phakic and remaining 3 eyes were pseudophakic. Among the 13 eyes which had false positives/negatives using the digital fundus photographs, 10 eyes were phakic and remaining 3 eyes were pseudophakic. Proportion of phakic or pseudophakic in each imaging modalities is shown in Table 6. Although there were no significant differences in three imaging modalities, ClarusTM tended to have less proportion of phakic eyes which showed false positives/negatives. Also, true positives/negatives were recorded in all pseudophakic eyes using the ClarusTM (Table 6).

7.Title has to be revised since this study only investigated neovascular AMD (nAMD). similar changes have to be made in introduction of abstract and manuscript.

Response: Thank you for your comment. We changed the title and also some changes were made in the manuscript.

Title

Sensitivity and Specificity of High-resolution Wide Field Fundus Imaging for Detecting Neovascular Age-related Macular Degeneration

Short title

Accuracy of ClarusTM for Neovascular AMD

Page 5, line 83-84

The purpose of this study was to evaluate the accuracy of ClarusTM, Optos®, and digital fundus photographs for screening neovascular AMD.

8. Abstract Conclusion : ‘improving therapeutic outcomes and maintenance of better visual acuity’ this is not supported by the study’s results. Similar changes needed in manuscripts conclusion line 264.

Response: Thank you for your comment. We deleted the words ‘improving therapeutic outcomes and maintenance of better visual acuity’ in the abstract and manuscripts conclusion.

9. Abstract Conclusion: ’superior for examination of detailed macular lesions’ is not supported by study’s results. Extensive evaluation of a device in multiple aspects is needed before attempting to claim superiority and this study is limited in terms of methodology. also the authors did not evaluate multiple macular lesions herein.

Response: Thank you for your comment. We changed the words from ‘examination of detailed macular lesions’ to ‘diagnosing neovascular AMD’ in the abstract and manuscripts conclusion.

Page 3, line 39-40 and page 20, line 317-318

ClarusTM, possessing an ultra-wide field imaging system, was considered superior for diagnosing neovascular AMD with high sensitivity and specificity.

Reviewer #2

1. The “Introduction” section (lines 56-79) could be significantly strengthened. While I am aware of your scientific work and your writing style, I would strongly suggest you expand this section, review the literature but also add relevant references. In many statements I believe that a reference is missing such as in line 59 but also the 2nd and the 3rd paragraphs have no references at all. I would strongly encourage you to at least site a website in order to support the content when needed. In addition, in the 2nd sentence (line 57) you will also need to clarify that you are actually talking about neovascular AMD. This should also be added in your “Purpose” section (Abstract, first sentence) but I would suggest you add it also in your title, the short title and in the “Discussion” (e.g. lines 88, 190). While these statements might be true, you should support those based on other publications and maybe add some epidemiological data as well. The same applies for line 59 regarding screening. In general, this weakness is found throughout your manuscript.

Response: We thank the reviewer for this suggestion. We expanded the ‘Introduction’ 

section and added many references. In the 2nd sentence, ‘Purpose’ section, the short 

title, and ‘Discussion’ section, I changed from ‘AMD’ to ‘neovascular AMD’. Finally, I 

listed 25 references.

2. In regard to your “Data Collection”, you do not really talk about the actual numbers. It seems like most of your patients had bilateral neovascular AMD. This is an important point not only to highlight it in your Methods but also consider it during your analysis and interpretation of the results.

Please, add a citation for ARM (lines 85-97).

Response: I’m sorry for confusing you. I clarified the number of neovascualr AMD in the data collection section. Also, I added the citation for ARM. 

Page 7, line 120-124

Of the 73 patients, 17 patients had bilateral neovascualr AMD, 37 fellow eyes showed age-related maculopathy (ARM), and remaining 19 fellow eyes were normal by the unmasked graders. Therefore, the fundus imaging of a consecutive series of 109 eyes of 73 patients (both treatment naïve and treated eyes) with neovascular AMD were retrospectively reviewed by the masked graders.

3. Line 107: When you say that “Data obtained from slit-lamp biomicroscopy and SD-OCT were reviewed by two unmasked graders” do you mean that the patient was examined by two ophthalmologists? What kind of data did you collect? Please, explain/describe those characteristics.

Response: Thank you for your comments. I described how to diagnose neovascualr AMD more clearly.

Page 7, line 110-112

The patients were examined by two unmasked graders (MM and YK). The findings obtained from slit-lamp biomicroscopy and SD-OCT imaging in the macula were reviewed by the unmasked graders.

4. Line 149: You say that you tested 109 from 73 patients. Those 19 eyes that were marked normal are not actually “the remaining”. You need to explain whether you scanned in total both eyes of the 73 patients (n=146) and if yes, what happened to the remaining 18 eyes. Did all of these belong to the ARM group? Were they excluded for other reason (e.g scan quality)?

Response: We thank the reviewer for the comment. 37 fellow eyes showed ARM, therefore, we excluded these eyes.

Page 7, line 120-124

Of the 73 patients, 17 patients had bilateral neovascualr AMD, 37 fellow eyes showed age-related maculopathy (ARM), and remaining 19 fellow eyes were normal by the unmasked graders. Therefore, the fundus imaging of a consecutive series of 109 eyes of 73 patients (both treatment naïve and treated eyes) with neovascular AMD were retrospectively reviewed by the masked graders.

5. Please, expand your “Introduction” and “Discussion” sections and talk more about the strengths and the weaknesses of these three devices. Also, based on the current literature, make a comment on the use of this device in other retina diseases.

Response: Thank you for your suggestion. I talked more about three devices. Also, I described the use of these devices in DR in the discussion section.

Line 227-234

Recently, an ultra-wide-field (UWF) imaging has been widely used for screening or identification of retinal pathology. [18] For example, Optos UWF images improved detection of peripheral lesions in DR and enabled to lead more accurate classification of the disease.[19] Regarding of ClarusTM, Hirano et al compared the ClarusTM with Optos® in patients with DR and described that both systems were useful for assessing DR severity.[20] However, the comparison of an UWF imaging with conventional digital fundus camera for diagnosing macular lesions, especially neovascular AMD, remained unclear.

Line 272-275

Although the sensitivity of Optos® tended to have lower than digital fundus photographs, there were no significant difference between the two modalities, which was consistent with the previous report that there was a good agreement between grading digital fundus camera and Optos® in the macula.[21]

6. I would include the lens status of those patients not only in the demographic table but also consider it as a parameter in your statistical analysis. You mention that “Clarus may not be easily affected by cataracts” (lines 206-207). If so, it would be interesting to see the results of this analysis.

Response: Thank you for the suggestion. I added subgroup analysis in phakic and pseudophakic eyes and discussed it. 

Page 13, line 199-209 and table 6

Of the 109 eyes, false positives or false negatives were recorded using the ClarusTM in 7 eyes, the Optos® in 18 eyes, and the digital fundus photographs in 13 eyes. All 7 eyes which showed false positives/negatives using the ClarusTM were phakic eyes. Of the 18 eyes which showed false positives/negatives using the Optos®, 15 eyes were phakic and remaining 3 eyes were pseudophakic. Among the 13 eyes which had false positives/negatives using the digital fundus photographs, 10 eyes were phakic and remaining 3 eyes were pseudophakic. Proportion of phakic or pseudophakic in each imaging modalities is shown in Table 6. Although there were no significant differences in three imaging modalities, ClarusTM tended to have less proportion of phakic eyes which showed false positives/negatives. Also, true positives/negatives were recorded in all pseudophakic eyes using the ClarusTM (Table 6).

Page 16, line 248-250

In fact, ClarusTM tended to have the least proportion of phakic eyes which showed false positives/negatives in three imaging devices.

7. Please, consult a statistician. Explain why you chose to use McNemar’s test and not Cohen’s.

Response: Cohen's kappa coefficient is a statistic which measures inter-rater agreement for qualitative (categorical) items. On the other hand, McNemar's test assesses the dependence of categorical data that are matched or paired. In this case, we think McNemar’s test is appropriate way for comparing the ability to diagnose neovascualr AMD by using paired data. 

8. Also, it would be better to use tables in order to present the results of your analysis.

Response: I appreciate your comment. I added table 2-6 to show the results.

Minor remarks

9. Consider adding keywords. This can increase the number of people finding and reading your article.

Response: Thank you for your comment. We added keywords.

10. You could make a comment on the male to female ratio of your cohort compared to other cohorts and studies.

Response: I appreciate your comment. I added the comment about male/female ratio.

Page 19, line 299-303

In this study, the male ratio was much higher than that of female. In Japan, patients with neovascular AMD is a male preponderance [22], which is converse to population-based studies in Caucasion.[23,24] Although the reason has not been clarified, genetic differences between Japanese and Caucasian patients or the higher smoking rate in male in Japanese people may underlie.

11. Line 252: please, review the sentence. In general, the quality of your manuscript can be improved even more after a careful proofreading.

Response: Thank you for your comment. We corrected the sentence.

Page 19, line 306-307

Although ClarusTM and Optos® can be usually taken without mydriasis, in this study, all images were taken with mydriasis.

---

## [Decision Letter · Decision Letter 1]

1 Jul 2020

PONE-D-20-09112R1

Sensitivity and Specificity of High-resolution Wide Field Fundus Imaging for Detecting Neovascular Age-related Macular Degeneration

PLOS ONE

Dear Dr. Maruyama-Inoue,

Thank you for submitting your manuscript to PLOS ONE. After careful consideration, we feel that it has merit but does not fully meet PLOS ONE’s publication criteria as it currently stands. Therefore, we invite you to submit a revised version of the manuscript that addresses the points raised during the review process.

Thank you for submitting your revised manuscript. Since there is no professional editing step before acceptance please make some more grammatical syntax edits before the final acceptance.  

Line 236: please, add an explanation about the abilities of the graders. In addition, there are several typos, grammar and syntactic errors such as in lines: 59, 112, 120, 141, 143, 152, 154, 193, 195, 207, 222, 226, 273, 276, 279, 300, 302, 318, 319. Please, revise.

We look forward to receiving your revised manuscript.

Kind regards,

Demetrios G. Vavvas

Academic Editor

PLOS ONE

Reviewers' comments:

Reviewer's Responses to Questions

**Comments to the Author**

1. If the authors have adequately addressed your comments raised in a previous round of review and you feel that this manuscript is now acceptable for publication, you may indicate that here to bypass the “Comments to the Author” section, enter your conflict of interest statement in the “Confidential to Editor” section, and submit your "Accept" recommendation.

Reviewer #2: All comments have been addressed

2. Is the manuscript technically sound, and do the data support the conclusions?

Reviewer #2: Yes

3. Has the statistical analysis been performed appropriately and rigorously? 

Reviewer #2: Yes

4. Have the authors made all data underlying the findings in their manuscript fully available?

Reviewer #2: Yes

5. Is the manuscript presented in an intelligible fashion and written in standard English?

Reviewer #2: Yes

6. Review Comments to the Author

Reviewer #2: Thank you for submitting your revised manuscript.

Please, find a few more comments and suggestions.

I would recommend instead of using the word “screening” to use “early detection” or “detecting” throughout your manuscript. In your conclusions, you could probably state whether this could be or not a promising screening tool.

Also, you could use the word “classified” instead of “judged” when referred to the graders.

Line 236: please, add a couple of abilities of the graders.

In addition, there are several typos, grammar and syntactic errors such as in lines: 59, 112, 120, 141, 143, 152, 154, 193, 195, 207, 222, 226, 273, 276, 279, 300, 302, 318, 319. Please, revise.

Thank you.

7. PLOS authors have the option to publish the peer review history of their article (what does this mean?). If published, this will include your full peer review and any attached files.

Reviewer #2: No

---

## [Author Response · Author response to Decision Letter 1]

11 Jul 2020

July 11th, 2020

Demetrios G. Vavvas, MD, PhD. 

Academic Editor

PLOS ONE

RE: PONE-D-20-09112, entitled "Sensitivity and Specificity of High-resolution Wide Field Fundus Imaging for Detecting Age-related Macular Degeneration"

Dear Dr. Demetrios G. Vavvas

Thank you very much for your kind reviewing our manuscript.

You will find the author’s comments for manuscript on this letter which was revised based on the reviewer’s comments.

We are looking forward to seeing your favorable comments in the future.

We greatly appreciate your editing work. 

With Best regards,

Corresponding author;

Maiko Maruyama-Inoue, M.D. 

Yokohama City University Medical Center

4-57 Urafune-cho, Minami-ku, Yokohama, Kanagawa 232-0024, Japan.

E-mail: maicoo@urahp.yokohama-cu.ac.jp

Fax: +81 45 253 8490

Phone: +81 45 261 5656 

Journal Requirements

1) Thank you for submitting your revised manuscript. Since there is no professional editing step before acceptance please make some more grammatical syntax edits before the final acceptance. 

Response: Thank you for your requirements. I made grammatical syntax edits.

Reviewer’s comments

Reviewer #2

1. I would recommend instead of using the word “screening” to use “early detection” or “detecting” throughout your manuscript. 

Response: We thank the reviewer for the suggestion. I changed the word ‘screening’ to ‘early detection’ or ‘detecting’. 

Line 61, line74, line83, line224, line312

2. In your conclusions, you could probably state whether this could be or not a promising screening tool.

Response: Thank you for the comment. I added the sentence in conclusions.

line 42-43, line 319-321 

It may be a useful screening tool for early detection of neovascular AMD, facilitating prompt referral and treatment.

3. Also, you could use the word “classified” instead of “judged” when referred to the graders.

Response: We thank the reviewer for the suggestion. I changed the word ‘judged’ to ‘classified’.

line 264, line 266, line 268, line 292, line 293, line 295

4. line 236: please, add a couple of abilities of the graders.

Response: Thank you for the comment. I described more detail about the graders.

Page 15, line 233-235,

In this study, two retina specialists, who had ophthalmology residency for more than 5 years and retina-vitreous fellowship for more than 2 years, were selected as masked graders with similar abilities as unmasked graders.

5. In addition, there are several typos, grammar and syntactic errors such as in lines: 59, 112, 120, 141, 143, 152, 154, 193, 195, 207, 222, 226, 273, 276, 279, 300, 302, 318, 319. Please, revise.

Response: Thank you for the comment. I revised several errors.

---

## [Editor Report · Decision Letter 2]

10 Aug 2020

Sensitivity and Specificity of High-resolution Wide Field Fundus Imaging for Detecting Neovascular Age-related Macular Degeneration

PONE-D-20-09112R2

Dear Dr. Maruyama-Inoue,

We’re pleased to inform you that your manuscript has been judged scientifically suitable for publication and will be formally accepted for publication once it meets all outstanding technical requirements.

Kind regards,

Demetrios G. Vavvas

Academic Editor

PLOS ONE
---

## [Editor Report · Acceptance letter]

12 Aug 2020

PONE-D-20-09112R2 

Sensitivity and Specificity of High-resolution Wide Field Fundus Imaging for Detecting Neovascular Age-related Macular Degeneration 

Dear Dr. Maruyama-Inoue:

I'm pleased to inform you that your manuscript has been deemed suitable for publication in PLOS ONE. Congratulations! Your manuscript is now with our production department. 

Kind regards, 

on behalf of

Dr. Demetrios G. Vavvas 

Academic Editor

PLOS ONE